# Testing for Differences in Gaussian Graphical Models: Applications to Brain Connectivity

**Eugene Belilovsky [1,2,3], Gael Varoquaux[2], Matthew Blaschko[3]**
[1]University of Paris-Saclay, [2]INRIA, [3]KU Leuven
{eugene.belilovsky, gael.varoquaux } @inria.fr
matthew.blaschko@esat.kuleuven.be

## Abstract

Functional brain networks are well described and estimated from data with Gaussian Graphical Models (GGMs), e.g. using sparse inverse covariance estimators. Comparing functional connectivity of subjects in two populations calls for comparing these estimated GGMs. Our goal is to identify differences in GGMs known to have similar structure. We characterize the uncertainty of differences with confidence intervals obtained using a parametric distribution on parameters of a sparse estimator. Sparse penalties enable statistical guarantees and interpretable models even in high-dimensional and low-sample settings. Characterizing the distributions of sparse models is inherently challenging as the penalties produce a biased estimator. Recent work invokes the sparsity assumptions to effectively remove the bias from a sparse estimator such as the lasso. These distributions can be used to give confidence intervals on edges in GGMs, and by extension their differences. However, in the case of comparing GGMs, these estimators do not make use of any assumed joint structure among the GGMs. Inspired by priors from brain functional connectivity we derive the distribution of parameter differences under a joint penalty when parameters are known to be sparse in the difference. This leads us to introduce the debiased multi-task fused lasso, whose distribution can be characterized in an efficient manner. We then show how the debiased lasso and multi-task fused lasso can be used to obtain confidence intervals on edge differences in GGMs. We validate the techniques proposed on a set of synthetic examples as well as neuro-imaging dataset created for the study of autism.

## 1 Introduction

Gaussian graphical models describe well interactions in many real-world systems. For instance, correlations in brain activity reveal brain interactions between distant regions, a process know as *functional connectivity*. Functional connectivity is an interesting probe on brain mechanisms as it persists in the absence of tasks (the so-called "resting-state") and is thus applicable to study populations of impaired subjects, as in neurologic or psychiatric diseases [3]. From a formal standpoint, Gaussian graphical models are well suited to estimate brain connections from functional Magnetic Resonance Imaging (fMRI) signals [28, 33]. A set of brain regions and related functional connections is then called a functional *connectome* [31, 3]. Its variation across subjects can capture cognition [26, 27] or pathology [17, 3]. However, the effects of pathologies are often very small, as resting-state fMRI is a weakly-constrained and noisy imaging modality, and the number of subjects in a study is often small given the cost of imaging. Statistical power is then a major concern [2]. The statistical challenge is to increase the power to detect differences between Gaussian graphical models in the small-sample regime.

In these settings, estimation and comparison of Gaussian graphical models fall in the range of high-dimensional statistics: the number of degrees of freedom in the data is small compared to

the dimensionality of the model. In this regime, sparsity-promoting $\ell_1$-based penalties can make estimation well-posed and recover good estimation performance despite the scarcity of data [29, 10, 22, 6, 1]. These encompass sparse regression methods such as the lasso or recovery methods such as basis pursuit, and can be applied to estimation of Gaussian graphical models with approaches such as the graphical lasso[10]. There is now a wide body of literature which demonstrates the statistical properties of these methods [1]. Crucial to applications in medicine or neuroscience, recent work characterizes the uncertainty, with confidence intervals and $p$-values, of the parameters selected by these methods [15, 16, 19, 12]. These works focus primarily on the lasso and graphical lasso.

Approaches to estimate statistical significance on sparse models fall into several general categories: (a) non-parameteric sampling based methods which are inherently expensive and have difficult limiting distributions [1, 24, 5], (b) characterizations of the distribution of new parameters that enter a model along a regularization path [19, 12], or (c) for a particular regularization parameter, debiasing the solution to obtain a new consistent estimator with known distribution [16, 15, 30]. While some of the latter work has been used to characterize confidence intervals on network edge selection, there is no result, to our knowledge, on the important problem of identifying differences in networks. Here the confidence on the result is even more critical, as the differences are the direct outcome used for neuroscience research or medical practice, and it is important to provide the practitioner a measure of the uncertainty.

Here, we consider the setting of two datasets known to have very similar underlying signals, but which individually may not be very sparse. A motivating example is determining the difference in brain networks of subjects from different groups: population analysis of connectomes [31, 17]. Recent literature in neuroscience [20] has suggested functional networks are not sparse. On the other hand, differences in connections across subjects should be sparse. Indeed the link between functional and anatomical brain networks [13] suggests they should not differ drastically from one subject to another. From a neuroscientific standpoint we are interested in determining which edges between two populations (e.g. autistic and non-autistic) are different. Furthermore we want to provide confidence-intervals on our results. We particularly focus on the setting where one dataset is larger than the other. In many applications it is more difficult to collect one group (e.g. individuals with specific pathologies) than another.

We introduce an estimator tailored to this goal: the debiased multi-task fused lasso. We show that, when the underlying parameter differences are indeed sparse, we can obtain a tractable Gaussian distribution for the parameter difference. This closed-form distribution underpins accurate hypothesis testing and confidence intervals. We then use the relationship between nodewise regression and the inverse covariance matrix to apply our estimator to learning differences of Gaussian graphical models.

The paper is organized as follows. In Section 2 we review previous work on learning of GGMs and the debiased lasso. Section 3 discusses a joint debiasing procedure that specifically debiases the difference estimator. In Section 3.1 we introduce the debiased multi-task fused lasso and show how it can be used to learn parameter differences in linear models. In Section 3.2, we show how these results can be used for GGMs. In Section 4 we validate our approach on synthetic and fMRI data.

## 2    Background and Related Work

**Debiased Lasso**    A central starting point for our work is the debiased lasso [30, 16]. Here one considers the linear regression model, $Y = \boldsymbol{X}\beta + \epsilon$, with data matrix $\boldsymbol{X}$ and output $Y$, corrupted by $\epsilon \sim N(0, \sigma_\epsilon^2 I)$ noise. The lasso estimator is formulated as follows:

$$\hat{\beta}^\lambda = \arg\min_\beta \frac{1}{n}\|Y - \boldsymbol{X}\beta\|^2 + \lambda\|\beta\|_1 \tag{1}$$

The KKT conditions give $\hat{k}^\lambda = \frac{1}{n}\boldsymbol{X}^T(Y - \boldsymbol{X}\beta)$, where $\hat{k}$ is the subgradient of $\lambda\|\beta\|_1$. The debiased lasso estimator [30, 16] is then formulated as $\hat{\beta}_u^\lambda = \hat{\beta}^\lambda + \boldsymbol{M}\hat{k}^\lambda$ for some $\boldsymbol{M}$ that is constructed to give guarantees on the asymptotic distribution of $\hat{\beta}_u^\lambda$. Note that this estimator is not strictly unbiased in the finite sample case, but has a bias that rapidly approaches zero (w.r.t. $n$) if $\boldsymbol{M}$ is chosen appropriately, the true regressor $\beta$ is indeed sparse, and the design matrix satistifes a certain restricted eigenvalue property [30, 16]. We decompose the difference of this debiased estimator and the truth as follows:

$$\hat{\beta}_u^\lambda - \beta = \frac{1}{n}\boldsymbol{M}\boldsymbol{X}^T\epsilon - (\boldsymbol{M}\hat{\Sigma} - I)(\hat{\beta} - \beta) \tag{2}$$

The first term is Gaussian and the second term is responsible for the bias. Using Holder's inequality the second term can be bounded by $\|M\hat{\Sigma} - I\|_\infty \|\hat{\beta} - \beta\|_1$. The first part of which we can bound using an appropriate selection of $M$ while the second part is bounded by our implicit sparsity assumptions coming from lasso theory [1]. Two approaches from the recent literature discuss how one can select $M$ to appropriately debias this estimate. In [30] it suffices to use nodewise regression to learn an inverse covariance matrix which guarantees constraints on $\|M\hat{\Sigma} - I\|_\infty$. A second approach by [16] proposes to solve a quadratic program to directly minimize the variance of the debiased estimator while constraining $\|M\hat{\Sigma} - I\|_\infty$ to induce sufficiently small bias.

Intuitively the construction of $\hat{\beta}_u^\lambda$ allows us to trade variance and bias via the $M$ matrix. This allows us to overcome a naive bias-variance tradeoff by leveraging the sparsity assumptions that bound $\|\hat{\beta} - \beta\|_1$. In the sequel we expand this idea to the case of debiased parameter difference estimates and sparsity assumptions on the parameter differences.

In the context of GGMs, the debiased lasso can gives us an estimator that asymptotically converges to the partial correlations. As highlighted by [34] we can thus use the debiased lasso to obtain difference estimators with known distributions. This allows us to obtain confidence intervals on edge differences between Gaussian graphical models. We discuss this further in the sequel.

**Gaussian Graphical Model Structure Learning**    A standard approach to estimating Gaussian graphical models in high dimensions is to assume sparsity of the precision matrix and have a constraint which limits the number of non-zero entries of the precision matrix. This constraint can be achieved with a $\ell_1$-norm regularizer as in the popular graphical lasso [10]. Many variants of this approach that incorporate further structural assumptions have been proposed [14, 6, 23].

An alternative solution to inducing sparsity on the precision matrix indirectly is neighborhood $\ell_1$ regression from [22]. Here the authors make use of a long known property that connects the entries of the precision matrix to the problem of regression of one variable on all the others [21]. This property is critical to our proposed estimation as it allows relating regression models to finding edges connected to specific nodes in the GGM.

GGMs have been found to be good at recovering the main brain networks from fMRI data [28, 33]. Yet, recent work in neuroscience has showed that the structural wiring of the brain did not correspond to a very sparse network [20], thus questioning the underlying assumption of sparsity often used to estimate brain network connectivity. On the other hand, for the problem of finding differences between networks in two populations, sparsity may be a valid assumption. It is well known that anatomical brain connections tend to closely follow functional ones [13]. Since anatomical networks do not differ drastically we can surmise that two brain networks should not differ much even in the presence of pathologies. The statistical method we present here leverages sparsity in the difference of two networks, to yield well-behaved estimation and hypothesis testing in the low-sample regime. Most closely related to our work, [35, 9] recently consider a different approach to estimating difference networks, but does not consider assigning significance to the detection of edges.

## 3   Debiased Difference Estimation

In many applications one may be interested in learning multiple linear models from data that share many parameters. Situations such as this arise often in neuroimaging and bioinformatics applications. We can often improve the learning procedure of such models by incorporating fused penalties that penalize the $\|\cdot\|_1$ norm of the parameter differences or $\|\cdot\|_{1,2}$ which encourages groups of parameters to shrink together. These methods have been shown to substantially improve the learning of the joint models. However, the differences between model parameters, which can have a high sample complexity when there are few of them, are often pointed out only in passing [4, 6, 14]. On the other hand, in many situations we might be interested in actually understanding and identifying the differences between elements of the support. For example when considering brain networks of patients suffering from a pathology and healthy control subjects, the difference in brain connectivity may be of great interest. Here we focus specifically on accurately identifying differences with significance.

We consider the case of two tasks (e.g. two groups of subjects), but the analysis can be easily extended to general multi-task settings. Consider the problem setting of data matrices $X_1$ and $X_2$, which are $n_1 \times p$ and $n_2 \times p$, respectively. We model them as producing outputs $Y_1$ and $Y_2$, corrupted by

diagonal gaussian noise $\epsilon_1$ and $\epsilon_2$ as follows

$$Y_1 = \boldsymbol{X_1}\beta_1 + \epsilon_1, \;\; Y_2 = \boldsymbol{X_2}\beta_2 + \epsilon_2 \tag{3}$$

Let $S_1$ and $S_2$ index the elements of the support of $\beta_1$ and $\beta_2$, respectively. Furthermore the support of $\beta_1 - \beta_2$ is indexed by $S_d$ and finally the union of $S_1$ and $S_2$ is denoted $S_a$. Using a squared loss estimator producing independent estimates $\hat{\beta}_1, \hat{\beta}_2$ we can obtain a difference estimate $\hat{\beta}_d = \hat{\beta}_1 - \hat{\beta}_2$. In general if $S_d$ is very small relative to $S_a$ then we will have a difficult time to identify the support $S_d$. This can be seen if we consider each of the individual components of the prediction errors. The larger the true support $S_a$ the more it will drown out the subset which corresponds to the difference support. This can be true even if one uses $\ell_1$ regularizers over the parameter vectors. Consequently, one cannot rely on the straightforward strategy of learning two independent estimates and taking their difference. The problem is particularly pronounced in the common setting where one group has fewer samples than the other. Thus here we consider the setting where $n_1 > n_2$ and possibly $n_1 \gg n_2$.

Let $\hat{\beta}_1$ and $\hat{\beta}_2$ be regularized least squares estimates. In our problem setting we wish to obtain confidence intervals on debiased versions of the difference $\hat{\beta}_d = \hat{\beta}_1 - \hat{\beta}_2$ in a high-dimensional setting (in the sense that $n_2 < p$), we aim to leverage assumptions about the form of the true $\beta_d$, primarily that it is sparse, while the independent $\hat{\beta}_1$ and $\hat{\beta}_2$ are weakly sparse or not sparse. We consider a general case of a joint regularized least squares estimation of $\hat{\beta}_1$ and $\hat{\beta}_2$

$$\min_{\beta_1, \beta_2} \frac{1}{n_1} \|Y_1 - \boldsymbol{X_1}\beta_1\|^2 + \frac{1}{n_2} \|Y_2 - \boldsymbol{X_2}\beta_2\|^2 + R(\beta_1, \beta_2) \tag{4}$$

We note that the differentiating and using the KKT conditions gives

$$\hat{k}^\lambda = \begin{bmatrix} \hat{k}_1 \\ \hat{k}_2 \end{bmatrix} = \begin{bmatrix} \frac{1}{n_1} \boldsymbol{X_1}^T (Y - \boldsymbol{X_1}\beta_1) \\ \frac{1}{n_2} \boldsymbol{X_2}^T (Y - \boldsymbol{X_2}\beta_2) \end{bmatrix} \tag{5}$$

where $\hat{k}^\lambda$ is the (sub)gradient of $R(\beta_1, \beta_2)$. Substituting Equation (3) we can now write

$$\hat{\Sigma}_1(\hat{\beta}_1 - \beta_1) + \hat{k}_1 = \frac{1}{n_1} \boldsymbol{X_1}^T \epsilon_1 \;\; \text{and} \;\; \hat{\Sigma}_2(\hat{\beta}_2 - \beta_2) + \hat{k}_2 = \frac{1}{n_2} \boldsymbol{X_2}^T \epsilon_2 \tag{6}$$

We would like to solve for the difference $\hat{\beta}_1 - \hat{\beta}_2$ but the covariance matrices may not be invertible. We introduce matrices $\boldsymbol{M}_1$ and $\boldsymbol{M}_2$, which will allow us to isolate the relevant term. We will see that in addition these matrices will allow us to decouple the bias and variance of the estimators.

$$\boldsymbol{M}_1\hat{\Sigma}_1(\hat{\beta}_1 - \beta_1) + \boldsymbol{M}_1\hat{k}_1 = \frac{1}{n_1} \boldsymbol{M}_1\boldsymbol{X_1}^T \epsilon_1 \;\; \text{and} \;\; \boldsymbol{M}_2\hat{\Sigma}_2(\hat{\beta}_2 - \beta_2) + \boldsymbol{M}_2\hat{k}_2 = \frac{1}{n_2} \boldsymbol{M}_2\boldsymbol{X_2}^T \epsilon_2 \tag{7}$$

subtracting these and rearranging we can now isolate the difference estimator plus a term we add back controlled by $\boldsymbol{M}_1$ and $\boldsymbol{M}_2$

$$(\hat{\beta}_1 - \hat{\beta}_2) - (\beta_1 - \beta_2) + \boldsymbol{M}_1\hat{k}_1 - \boldsymbol{M}_2\hat{k}_2 = \frac{1}{n_1} \boldsymbol{M}_1\boldsymbol{X_1}^T \epsilon_1 - \frac{1}{n_2} \boldsymbol{M}_2\boldsymbol{X_2}^T \epsilon_2 - \Delta \tag{8}$$

$$\Delta = (\boldsymbol{M}_1\hat{\Sigma}_1 - I)(\hat{\beta}_1 - \beta_1) - (\boldsymbol{M}_2\hat{\Sigma}_2 - I)(\hat{\beta}_2 - \beta_2) \tag{9}$$

Denoting $\beta_d := \beta_1 - \beta_2$ and $\beta_a := \beta_1 + \beta_2$, we can reformulate $\Delta$:

$$\Delta = \frac{(\boldsymbol{M}_1\hat{\Sigma}_1 - I + \boldsymbol{M}_2\hat{\Sigma}_2 - I)}{2}(\hat{\beta}_d - \beta_d) + \frac{(\boldsymbol{M}_1\hat{\Sigma}_1 - \boldsymbol{M}_2\hat{\Sigma}_2)}{2}(\hat{\beta}_a - \beta_a) \tag{10}$$

Here, $\Delta$ will control the bias of our estimator. Additionally, we want to minimize its variance,

$$\frac{1}{n_1} \boldsymbol{M}_1\hat{\Sigma}_1\boldsymbol{M}_1\hat{\sigma}_1^2 + \frac{1}{n_2} \boldsymbol{M}_2\hat{\Sigma}_2\boldsymbol{M}_2\hat{\sigma}_2^2. \tag{11}$$

We can now overcome the limitations of simple bias variance trade-off by using an appropriate regularizer coupled with an assumption on the underlying signal $\beta_1$ and $\beta_2$. This will in turn make $\Delta$ asymptotically vanish while maximizing the variance.

Since we are interested in pointwise estimates, we can focus on bounding the infinity norm of $\Delta$.

$$\|\Delta\|_\infty \leq \frac{1}{2} \underbrace{\|\boldsymbol{M}_1\hat{\Sigma}_1 + \boldsymbol{M}_2\hat{\Sigma}_2 - 2I\|_\infty}_{\mu_1} \underbrace{\|\hat{\beta}_d - \beta_d\|_1}_{l_d} + \frac{1}{2} \underbrace{\|\boldsymbol{M}_1\hat{\Sigma}_1 - \boldsymbol{M}_2\hat{\Sigma}_2\|_\infty}_{\mu_2} \underbrace{\|\hat{\beta}_a - \beta_a\|_1}_{l_a} \tag{12}$$

We can control the maximum bias by selecting $M_1$ and $M_2$ appropriately. If we use an appropriate regularizer coupled with sparsity assumptions we can bound the terms $l_a$ and $l_d$ and use this knowledge to appropriately select $M_1$ and $M_2$ such that the bias becomes neglibile. If we had only the independent parameter sparsity assumption we can apply the results of the debiased lasso and estimate $M_1$ and $M_2$ independently as in [16]. In the case of interest where $\beta_1$ and $\beta_2$ share many weights we can do better by taking this as an assumption and applying a sparsity regularization on the difference by adding the term $\lambda_2\|\beta_1 - \beta_2\|_1$. Comparing the decoupled penalty to the fused penalty proposed we see that $l_d$ would decrease at a given sample size. We now show how to jointly estimate $M_1$ and $M_2$ so that $\|\Delta\|_\infty$ becomes negligible for a given $n$, $p$ and sparsity assumption.

## 3.1   Debiasing the Multi-Task Fused Lasso

Motivated by the inductive hypothesis from neuroscience described above we introduce a consistent low-variance estimator, the debiased multi-task fused lasso. We propose to use the following regularizer $R(\beta_1, \beta_2) = \lambda_1\|\beta_1\|_1 + \lambda_1\|\beta_2\|_1 + \lambda_2\|\beta_1 - \beta_2\|_1$. This penalty has been referred to in some literature as the multi-task fused lasso [4]. We propose to then debias this estimate as shown in (8). We estimate the $M_1$ and $M_2$ matrices by solving the following QP for each row $m_1$ and $m_2$ of the matrices $M_1$ and $M_2$.

$$\min_{m_1,m_2} \frac{1}{n_1} m_1^T \hat{\Sigma}_1 m_1 + \frac{1}{n_2} m_2^T \hat{\Sigma}_2 m_2 \tag{13}$$

$$s.t. \quad \|M_1\hat{\Sigma}_1 + M_2\hat{\Sigma}_2 - 2I\|_\infty \leq \mu_1, \quad \|M_1\hat{\Sigma}_1 - M_2\hat{\Sigma}_2\|_\infty \leq \mu_2$$

This directly minimizes the variance, while bounding the bias in the constraint. We now show how to set the bounds:

**Proposition 1.** *Take $\lambda_1 > 2\sqrt{\frac{\log p}{n_2}}$ and $\lambda_2 = O(\lambda_1)$. Denote $s_d$ the difference sparsity, $s_{1,2}$ the parameter sparsity $|S_1| + |S_2|$, $c > 1, a > 1$, and $0 < m \ll 1$. When the compatibility condition [1, 11] holds the following bounds gives $l_a u_2 = o(1)$ and $l_d u_1 = o(1)$ and thus $\|\Delta\|_\infty = o(1)$ with high probability.*

$$\mu_1 \leq \frac{1}{c\lambda_2 s_d n_2^m} \quad and \quad \mu_2 \leq \frac{1}{a(\lambda_1 s_{1,2} + \lambda_2 s_d)n_2^m} \tag{14}$$

The proof is given in the supplementary material. Using the prescribed $M$s obtained with (13) and 14 we obtain an unbiased estimator given by (8) with variance (11)

## 3.2   GGM Difference Structure Discovery with Significance

The debiased lasso and the debiased multi-task fused lasso, proposed in the previous section, can be used to learn the structure of a difference of Gaussian graphical models and to provide significance results on the presence of edges within the difference graph. We refer to these two procedures as Difference of Neighborhoods Debiased Lasso Selection and Difference of Neighborhoods Debiased Fused Lasso Selection.

We recall that the conditional independence properties of a GGM are given by the zeros of the precision matrix and these zeros correspond to the zeros of regression parameters when regressing one variable on all the other. By obtaining a debiased lasso estimate for each node in the graph [34] notes this leads to a sparse unbiased precision matrix estimate with a known asymptotic distribution. Subtracting these estimates for two different datasets gives us a difference estimate whose zeros correspond to no difference of graph edges in two GGMs. We can similarly use the debiased multi-task fused lasso described above and the joint debiasing procedure to obtain a test statistic for the difference of networks. We now formalize this procedure.

**Notation**   Given GGMs $j = 1, 2$. Let $X_j$ denote the random variable in $\mathbb{R}^p$ associated with GGM $j$. We denote $X_{j,v}$ the random variable associated with a node, $v$ of the GGM and $X_{j,v^c}$ all other nodes in the graph. We denote $\hat{\beta}_{j,v}$ the lasso or multi-task fused lasso estimate of $X_{j,v^c}$ onto $X_{j,v}$, then $\hat{\beta}_{j,dL,v}$ is the debiased version of $\hat{\beta}_{j,v}$. Finally let $\beta_{j,v}$ denote the unknown regression, $X_{j,v} = X_{j,v^c}\beta_{j,v} + \epsilon_j$ where $\epsilon_j \sim \mathbf{N}(0, \sigma_j\mathbf{I})$. Define $\beta_{D,v}^i = \hat{\beta}_{1,dL,v}^i - \hat{\beta}_{2,dL,v}^i$ the test statistic associated with the edge $v, i$ in the difference of GGMs $j = 1, 2$.

| **Algorithm 1** Difference Network Selection with Neighborhood Debiased Lasso | **Algorithm 2** Difference Network Selection with Neighborhood Debiased Fused Lasso |
|---|---|
| $V = \{1, ..., P\}$ | $V = \{1, ..., P\}$ |
| NxP Data Matrices, $\boldsymbol{X}_1$ and $\boldsymbol{X}_2$ | NxP Data Matrices, $\boldsymbol{X}_1$ and $\boldsymbol{X}_2$ |
| Px(P-1) Output Matrix $\boldsymbol{B}$ of test statistics | Px(P-1) Output Matrix $\boldsymbol{B}$ of test statistics |
| **for** $v \in V$ **do** | **for** $v \in V$ **do** |
|   Estimate unbiased $\hat{\sigma}_1, \hat{\sigma}_2$ from $X_{1,v}, X_{2,v}$ |   Estimate unbiased $\hat{\sigma}_1, \hat{\sigma}_2$ from $X_{1,v}, X_{2,v}$ |
|   **for** $j \in \{1, 2\}$ **do** |   $\beta_1, \beta_2 \leftarrow FusedLasso(\boldsymbol{X}_{1,v^c}, X_{1,v}, \boldsymbol{X}_{2,v^c}, X_{2,v})$ |
|     $\beta_j \leftarrow SolveLasso(\boldsymbol{X}_{j,v^c}, X_{j,v})$ |   $\boldsymbol{M}_1, \boldsymbol{M}_2 \leftarrow MEstimator(\boldsymbol{X}_{1,v^c}, \boldsymbol{X}_{2,v^c})$ |
|     $\boldsymbol{M}_j \leftarrow MEstimator(\boldsymbol{X}_{j,v^c})$ |   **for** $j \in \{1, 2\}$ **do** |
|     $\beta_{j,U} \leftarrow \beta_j + \boldsymbol{M}_j \boldsymbol{X}_{j,v^c}^T (X_{j,v} - \boldsymbol{X}_{j,v^c}\beta_j)$ |     $\beta_{j,U} \leftarrow \beta_j + \boldsymbol{M}_j \boldsymbol{X}_{j,v^c}^T (X_{j,v} - \boldsymbol{X}_{j,v^c}\beta_j)$ |
|   **end for** |   **end for** |
|   $\sigma_d^2 \leftarrow diag(\frac{\hat{\sigma}_1^2}{n_1}\boldsymbol{M}_1^T\hat{\Sigma}_1\boldsymbol{M}_1 + \frac{\hat{\sigma}_2^2}{n_2}\boldsymbol{M}_2^T\hat{\Sigma}_2\boldsymbol{M}_2)$ |   $\sigma_d^2 \leftarrow diag(\frac{\hat{\sigma}_1^2}{n_1}\boldsymbol{M}_1^T\hat{\Sigma}_1\boldsymbol{M}_1 + \frac{\hat{\sigma}_2^2}{n_2}\boldsymbol{M}_2^T\hat{\Sigma}_2\boldsymbol{M}_2)$ |
|   **for** $j \in v^c$ **do** |   **for** $j \in v^c$ **do** |
|     $\boldsymbol{B}_{v,j} = (\beta_{1,U,j} - \beta_{2,U,j})/\sqrt{\sigma_{d,j}^2}$ |     $\boldsymbol{B}_{v,j} = (\beta_{1,U,j} - \beta_{2,U,j})/\sqrt{\sigma_{d,j}^2}$ |
|   **end for** |   **end for** |
| **end for** | **end for** |

**Proposition 2.** *Given the $\hat{\beta}_{D,v}^i$, $\boldsymbol{M}_1$ and $\boldsymbol{M}_2$ computed as in [16] for the debiased lasso or as in Section 3.1 for the debiased multi-task fused lasso. When the respective assumptions of these estimators are satisfied the following holds w.h.p.*

$$\hat{\beta}_{D,v}^i - \beta_{D,v}^i = W + o(1) \ \ where \ \ W \sim \mathbf{N}(0, [\sigma_1^2\boldsymbol{M}_1\hat{\Sigma}_1\boldsymbol{M}_1^T + \sigma_2^2\boldsymbol{M}_2\hat{\Sigma}_2\boldsymbol{M}_2^T]_{i,i}) \qquad (15)$$

This follows directly from the asymptotic consistency of each individual $\hat{\beta}_{j,dL,v}^i$ for the debiased lasso and multi-task fused lasso.

We can now define the the null hypothesis of interest as $H_0 : \boldsymbol{\Theta}_{1,(i,j)} = \boldsymbol{\Theta}_{2,(i,j)}$. Obtaining a test statistic for each element $\beta_{D,v}^i$ allows us to perform hypothesis testing on individual edges, all the edges, or groups of edges (controlling for the FWER). We summarize the Neighbourhood Debiased Lasso Selection process in Algorithm 1 and the Neighbourhood Debiased Multi-Task Fused Lasso Selection in Algorithm 2 which can be used to obtain a matrix of all the relevant test statistics.

## 4 Experiments

### 4.1 Simulations

We generate synthetic data based on two Gaussian graphical models with 75 vertices. Each of the individual graphs have a sparsity of 19% and their difference sparsity is 3%. We construct the models by taking two identical precision matrices and randomly removing some edges from both. We generate synthetic data using both precision matrices. We use $n_1 = 800$ samples for the first dataset and vary the second dataset $n_2 = 20, 30, ...150$.

We perform a regression using the debiased lasso and the debiased multi-task fused lasso on each node of the graphs. As an extra baseline we consider the projected ridge method from the R package "hdi" [7]. We use the debiased lasso of [16] where we set $\lambda = k\hat{\sigma}\sqrt{\log p/n}$. We select $c$ by 3-fold cross validation $k = \{0.1, ..100\}$ and $\boldsymbol{M}$ as prescribed in [16] which we obtain by solving a quadratic program. $\hat{\sigma}$ is an unbiased estimator of the noise variance. For the debiased lasso we let both $\lambda_1 = k_1\hat{\sigma}_2\sqrt{\log p/n_2}$ and $\lambda_2 = k_2\hat{\sigma}_2\sqrt{\log p/n_2}$, and select based on 3-fold cross-validation from the same range as $k$. $\boldsymbol{M}_1$ and $\boldsymbol{M}_2$ are obtained as in Equation (13) with the bounds (14) being set with $c = a = 2$, $s_d = 2, s_{1,2} = 15, m = 0.01$, and the cross validated $\lambda_1$ and $\lambda_2$. In both debiased lasso and fused multi-task lasso cases we utilize the Mosek QP solver package to obtain $\boldsymbol{M}$. For the projected ridge method we use the hdi package to obtain two estimates of $\beta_1$ and $\beta_2$ along with their upper bounded biases which are then used to obtain $p$-values for the difference.

We report the false positive rate, the power, the coverage and interval length as per [30] for the difference of graphs. In these experiments we aggregate statistics to demonstrate power of the test statistic, as such we consider each edge as a separate test and do not perform corrections. Table 1 gives the numerical results for $n_2 = 60$: the power and coverage is substantially better for the debiased fused multi-task lasso, while at the same time the confidence interval smaller.

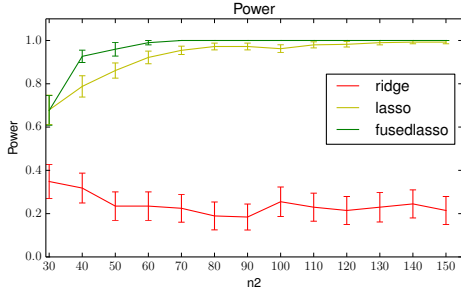

**Figure 1:** Power of the test for different number of samples in the second simulation, with $n_1 = 800$. The debiased fused lasso has highest statistical power.

| Method | FP | TP(Power) | Cov $S$ | Cov $S_d^c$ | len $S$ | len $S_d^c$ |
|---|---|---|---|---|---|---|
| Deb. Lasso | 3.7% | 80.6% | 96.2% | 92% | 2.199 | 2.195 |
| Deb. Fused Lasso | 0.0% | 93.3% | 100% | 98.6% | 2.191 | 2.041 |
| Ridge Projection | 0.0% | 18.6% | 100% | 100% | 5.544 | 5.544 |

**Table 1:** Comparison of Debiased Lasso, Debiased Fused Lasso, and Projected Ridge Regression for edge selection in difference of GGM. The significance level is 5%, $n_1 = 800$ and $n_2 = 60$. All methods have false positive below the significance level and the debiased fused lasso dominates in terms of power. The coverage of the difference support and non-difference support is also best for the debiased fused lasso, which simultaneously has smaller confidence intervals on average.

Figure 1 shows the power of the test for different values of $n_2$. The fusedlasso outperforms the other methods substantially. Projected ridge regression is particularly weak, in this scenario, as it uses a worst case p-value obtained using an estimate of an upper bound on the bias [7].

### 4.2 Autism Dataset

Correlations in brain activity measured via fMRI reveal functional interactions between remote brain regions [18]. In population analysis, they are used to measure how connectivity varies between different groups. Such analysis of brain function is particularly important in psychiatric diseases, that have no known anatomical support: the brain functions in a pathological aspect, but nothing abnormal is clearly visible in the brain tissues. Autism spectrum disorder is a typical example of such ill-understood psychiatric disease. Resting-state fMRI is accumulated in an effort to shed light on this diseases mechanisms: comparing the connectivity of autism patients versus control subjects. The ABIDE (Autism Brain Imaging Data Exchange) dataset [8] gathers rest-fMRI from 1 112 subjects across, with 539 individuals suffering from autism spectrum disorder and 573 typical controls. We use the preprocessed and curated data[1].

In a connectome analysis [31, 26], each subject is described by a GGM measuring functional connectivity between a set of regions. We build a connectome from brain regions of interest based on a multi-subject atlas[2] of 39 functional regions derived from resting-state fMRI [32] (see. Fig. 4).

We are interested in determining edge differences between the autism group and the control group. We use this data to show how our parametric hypothesis test can be used to determine differences in brain networks. Since no ground truth exists for this problem, we use permutation testing to evaluate the statistical procedures [25, 5]. Here we permute the two conditions (e.g. autism and control group) to compute a p-value and compare it to our test statistics. This provides us with a finite sample strict control on the error rate: a non-parametric validation of our parametric test.

For our experiments we take 2000 randomly chosen volumes from the control group subjects and 100 volumes from the autism group subjects. We perform permutation testing using the de-biased lasso, de-biased multi-task fused lasso, and projected ridge regression. Parameters for the de-biased fused lasso are chosen as in the previous section. For the de-biased lasso we use the exact settings for $\lambda$ and constraints on $M$ provided in the experimental section of [16]. Projected ridge regression is evaluated as in the previous section.

Figure 2 shows a comparison of three parametric approaches versus their analogue obtained with a permutation test. The chart plots the permutation p-values of each entry in the $38 \times 39$ $\boldsymbol{B}$ matrix against the expected parametric p-value. For all the methods the points are above the line indicating the tests are not breaching the expected false positive rates. However the de-biased lasso and ridge projecting are very conservative and lead to few detections. The de-biased multi-task fused lasso yields far more detections on the same dataset, within the expected false positive rate or near it.

We now analyse the reproducibility of the results by repeatedly sampling 100 subsets of the data (with the same proportions $n_1 = 2000$ and $n_2 = 100$), obtaining the matrix of test statistics, selecting edges that fall below the 5% significance level. Figure 3 shows how often edges are selected multiple times across subsamples. We report results with a threshold on uncorrected p-values as the lasso procedure

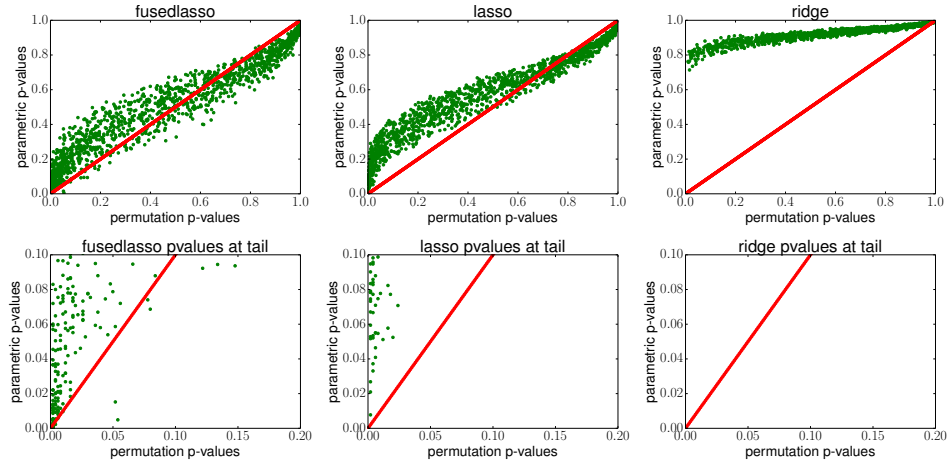

**Figure 2:** Permutation testing comparing debiased fused lasso, debiased lasso, and projected ridge regression on the ABIDE dataset. The chart plots the permutation p-values of each method on each possible edge against the expected parametric p-value. The debiased lasso and ridge projection are very conservative and lead to few detections. The fused lasso yields far more detections on the same dataset, almost all within the expected false positive rate.

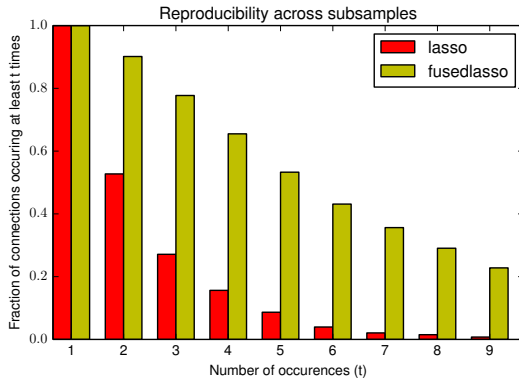

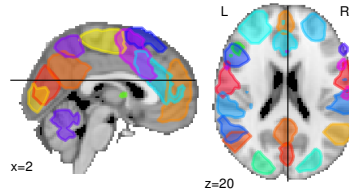

**Figure 4:** Outlines of the regions of the MSDL atlas.

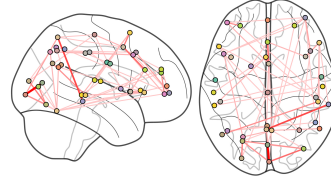

**Figure 3:** Reproducibility of results from sub-sampling using uncorrected error rate. The fused lasso is much more likely to detect edges and produce stable results. Using corrected p-values no detections are made by lasso (figure in supplementary material).

**Figure 5:** Connectome of repeatedly picked up edges in 100 trials. We only show edges selected more than once. Darker red indicates more frequent selection.

selects no edges with multiple comparison correction (supplementary materials give FDR-corrected results for the de-biased fused multi-task lasso selection). Figure 5 shows a connectome of the edges frequently selected by the de-biased fused multi-task lasso (with FDR correction).

## 5 Conclusions

We have shown how to characterize the distribution of differences of sparse estimators and how to use this distribution for confidence intervals and p-values on GGM network differences. For this purpose, we have introduced the de-biased multi-task fused lasso. We have demonstrated on synthetic and real data that this approach can provide accurate p-values and a sizable increase of statistical power compared to standard procedures. The settings match those of population analysis for functional brain connectivity, and the gain in statistical power is direly needed to tackle the low sample sizes [2].

Future work calls for expanding the analysis to cases with more than two groups as well as considering a $\ell_{1,2}$ penalty sometimes used at the group level [33]. Additionally the squared loss objective optimizes excessively the prediction and could be modified to lower further the sample complexity in terms of parameter estimation.

**Acknowledgements**

This work is partially funded by Internal Funds KU Leuven, ERC Grant 259112, FP7-MC-CIG 334380, and DIGITEO 2013-0788D - SOPRANO, and ANR-11-BINF-0004 NiConnect.

## Footnotes

[1]http://preprocessed-connectomes-project.github.io/abide/

[2]https://team.inria.fr/parietal/research/spatial_patterns/spatial-patterns-in-resting-state/

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
