[Supplementary Material · main.10-12.pdf]

# A   Analysis of Debiased Multi-Task Fused Lasso

The following analysis is used to show the conditions under which the debias multi-task fused lasso achieves a negligible bias.

Let $\beta = [\beta_1; \beta_2], \beta_d = \beta_1 - \beta_2, \beta_a = \beta_1 + \beta_2$. Let $S_{1,2}$ bet the support of $\beta$. Define $\boldsymbol{X}_N = [\boldsymbol{X}_1/\sqrt{n_1}, 0; 0, \boldsymbol{X}_2/\sqrt{n_2}]$

**Lemma 1.** *(Basic Inequality)* $\|\boldsymbol{X}_N(\hat{\beta} - \beta)\|_2^2 + \lambda_1\|\hat{\beta}\|_1 + \lambda_2\|\hat{\beta}_d\| \leq 2\epsilon^T\boldsymbol{X}_N(\hat{\beta} - \beta) + \lambda_1\|\beta\|_1 + \lambda_2\|\beta_d\|_1$

This follows from the fact that $\hat{\beta}$ is the minimizer of the fused lasso objective.

The term, $\epsilon^T\boldsymbol{X}_N(\hat{\beta} - \beta)$, commonly known as the empirical process term [1] can be bound as follows:

$$2|\epsilon^T\boldsymbol{X}_N(\hat{\beta} - \beta)| = 2|\epsilon_1^T\boldsymbol{X}_1(\hat{\beta}_1 - \beta_1)/n_1 + \epsilon_2^T\boldsymbol{X}_2(\hat{\beta}_2 - \beta_2)/n_2| \leq$$

$$2\|\hat{\beta}_1 - \beta_1\|_1 \max_{1 \leq j \leq p} |\epsilon_1^T\boldsymbol{X}_1^{(j)}/n_1| + 2\|\hat{\beta}_2 - \beta_2\|_1 \max_{1 \leq j \leq p} |\epsilon_2^T\boldsymbol{X}_2^{(j)}/n_2|$$

Where we utilize holder's inequality in the last line. We define the random event $\mathcal{F}$ for which the following holds: $\max_{1 \leq j \leq p} |\epsilon_1^T\boldsymbol{X}_1^{(j)}/n_1| \leq \lambda_0$ and $\max_{1 \leq j \leq p} |\epsilon_1^T\boldsymbol{X}_1^{(j)}/n_1| \leq \lambda_0$. furthermore we can select $2\lambda_0 \leq \lambda_1$

**Lemma 2.** *Suppose $\hat{\Sigma}_{j,j} = 1$ for both $\boldsymbol{X}_1$ and $\boldsymbol{X}_2$ then we have for all $t > 0$ and $n_1 > n_2$*

$$\lambda_0 = 2\sigma_2\sqrt{\frac{t^2 + \log p}{n_2}} \tag{16}$$

$$P(\mathcal{F}) = 1 - 2\exp(-t^2/2) \tag{17}$$

*Proof.* This follows directly from the [1, Lemma 6.2] and taking $n_1 > n_2$. $\qquad\square$

This allows us to get rid of the empirical process term on $\mathcal{F}$, with an appropriate choice of $\lambda_1$.

Given a set, $S$, denote $\beta_S$ the vector of equal size to $\beta$ but all elements not in $S$ set to zero. We can now show the following

**Lemma 3.** *We have on $\mathcal{F}$ with $\lambda_1 \geq 2\lambda_0$*

$$2\|\boldsymbol{X}_N(\hat{\beta} - \beta)\|_2^2 + \lambda_1\|\hat{\beta}_{S_{1,2}^c}\|_1 + 2\lambda_2\|\hat{\beta}_{d,S_d^c}\|_1$$

$$\leq 3\lambda_1\|\hat{\beta}_{S_{1,2}} - \beta_{S_{1,2}}\|_1 + 2\lambda_2\|\hat{\beta}_{d,S_d} - \beta_{d,S_d}\|_1 \tag{18}$$

*Proof.* Following [1, Lemma 6.3] we start with the basic inequality on $\mathcal{F}$. Which gives

$$2\|\boldsymbol{X}_N(\hat{\beta} - \beta)\|_2^2 + 2\lambda_1\|\hat{\beta}\|_1 + 2\lambda_2\|\hat{\beta}_d\|$$

$$\leq \lambda_1\|\hat{\beta} - \beta\|_1 + 2\lambda_1\|\beta\|_1 + 2\lambda_2\|\beta_d\|_1 \tag{19}$$

Since we assume the truth is in fact sparse,

$$\|\hat{\beta}_d - \beta_d\|_1 = \|\hat{\beta}_{d,S_d} - \beta_{d,S_d}\|_1 + \|\hat{\beta}_{d,S_d^C}\|_1 \tag{20}$$

$$\|\hat{\beta} - \beta\|_1 = \|\hat{\beta}_{S_{1,2}} - \beta_{S_{1,2}}\|_1 + \|\hat{\beta}_{S_{1,2}^C}\|_1 \tag{21}$$

Furthermore,

$$\|\hat{\beta}\|_1 \geq \|\beta_{S_{1,2}}\|_1 - \|\hat{\beta}_{S_{1,2}} - \beta_{S_{1,2}}\|_1 + \|\hat{\beta}_{S_{1,2}^C}\|_1 \tag{22}$$

$$\|\hat{\beta}_d\|_1 \geq \|\beta_{d,S_d}\|_1 - \|\hat{\beta}_{d,S_d} - \beta_{d,S_d}\|_1 + \|\hat{\beta}_{d,S_d^C}\|_1 \tag{23}$$

Substituting (22), (23), and (21) into (19) and rearranging completes the proof. $\qquad\square$

From the lemma above we can now justify the bounds in (14)

**Proposition 3.** *Take $\lambda_1 > 2\sqrt{\frac{\log p}{n_2}}$ and $\lambda_2 = O(\lambda_1)$. Denote $s_d$ the difference sparsity, $s_{1,2}$ the parameter sparsity $|S_1| + |S_2|$, $c > 1, a > 1$, and $0 < m \ll 1$. When the compatibility condition [1, 11] holds the following bounds gives $l_a u_2 = o(1)$ and $l_d u_1 = o(1)$ and thus $\|\Delta\|_\infty = o(1)$ with high probability.*

$$\mu_1 \leq \frac{1}{c\lambda_2 s_d n_2^m} \quad and \quad \mu_2 \leq \frac{1}{a(\lambda_1 s_{1,2} + \lambda_2 s_d)n_2^m} \tag{24}$$

*Proof.* We first consider the bound associated with $l_a$

$$\lambda_1 \|\hat{\beta}_a - \beta_a\|_1 \leq \lambda_1 \|\hat{\beta}_{S_{1,2}} - \beta_{S_{1,2}}\|_1 + \lambda_1 \|\hat{\beta}_{S_{1,2}^c}\|_1 \leq$$

$$4\lambda_1 \|\hat{\beta}_{S_{1,2}} - \beta_{S_{1,2}}\|_1 + 2\lambda_2 \|\hat{\beta}_{S_d} - \beta_{S_d}\|_1 - 2\|\boldsymbol{X}_N(\hat{\beta} - \beta)\|_2^2 \tag{25}$$

$$\leq 4\lambda_1 \sqrt{s_{1,2}} \|\hat{\beta}_{S_{1,2}} - \beta_{S_{1,2}}\|_2 + 2\lambda_2 \sqrt{s_d} \|\hat{\beta}_{S_d} - \beta_{S_d}\|_2$$
$$-2\|\boldsymbol{X}_N(\hat{\beta} - \beta)\|_2^2 \tag{26}$$

Invoking the compatibility assumption [1, 16, 11] with compatibility constant $\phi_{\min}$

$$\leq \frac{4\lambda_1 \sqrt{s_{1,2}}}{\phi_{min}} \|\boldsymbol{X}_N(\hat{\beta} - \beta)\|_2 + \frac{2\lambda_2 \sqrt{s_d}}{\phi_{min}} \|\boldsymbol{X}_N(\hat{\beta} - \beta)\|_2$$
$$-2\|\boldsymbol{X}_N(\hat{\beta} - \beta)\|_2^2 \tag{27}$$

$$\leq \frac{4\lambda_1^2 s_{1,2}}{\phi_{min}^2} + \frac{2\lambda_2^2 s_d}{\phi_{min}^2} \tag{28}$$

The bound $u_2$ now follows by inverting the expression shown and adding a factor of $n_2^m$ where $m \ll 1$.

Now we consider the bound for $l_d$.

$$\lambda_2 \|\hat{\beta}_d - \beta_d\|_1 = \lambda_2 \|\hat{\beta}_{d,S} - \beta_{d,S}\|_1 + \lambda_2 \|\hat{\beta}_{d,S^c}\|_1 \tag{29}$$

$$\leq 2\lambda_2 \|\hat{\beta}_{d,S} - \beta_{d,S}\|_1 + 3\lambda_1 \|\hat{\beta}_{S_{1,2}} - \beta_{S_{1,2}}\|_1/2 \tag{30}$$

$$-\|\boldsymbol{X}_N(\hat{\beta} - \beta)\|_2^2 - \lambda_1 \|\hat{\beta}_{S_{1,2}^c}\|_1/2 \tag{31}$$

In the domain of interest $n_1 \gg n_2$ if we select $\lambda_2 = O(\lambda_1)$ we can see the relevant terms related to the parameter support become small with respect to terms with $S_{1,2}$. Thus the error on the difference should dominate. In this region we can have $3\lambda_1 \|\hat{\beta}_{S_{1,2}} - \beta_{S_{1,2}}\|_1/2 - \lambda_1 \|\hat{\beta}_{S_{1,2}^c}\|_1/2 \leq c\lambda_2 \|\hat{\beta}_{d,S} - \beta_{d,S}\|_1$ where $c > 0$.

$$\lambda_2 \|\hat{\beta}_d - \beta_d\|_1 \leq 2\lambda_2 \|\hat{\beta}_{d,S} - \beta_{d,S}\|_1 - \|\boldsymbol{X}_N(\hat{\beta} - \beta)\|_2^2 \tag{32}$$

$$\leq 2c\lambda_2 \sqrt{s_d} \|\hat{\beta}_{d,S} - \beta_{d,S}\|_2 - \|\boldsymbol{X}_N(\hat{\beta} - \beta)\|_2^2 \tag{33}$$

Invoking the compatibility assumption [1]

$$\leq 2c\lambda_2 \sqrt{s_d} \|\boldsymbol{X}_N(\hat{\beta} - \beta)\|_2/\phi_{min} - \|\boldsymbol{X}_N(\hat{\beta} - \beta)\|_2^2 \tag{34}$$

$$\leq \frac{c^2 \lambda_2^2 s_d}{\phi_{min}^2} + \|\boldsymbol{X}_N(\hat{\beta} - \beta)\|_2^2 - \|\boldsymbol{X}_N(\hat{\beta} - \beta)\|_2^2 \tag{35}$$

Thus $\|\hat{\beta}_d - \beta_d\|_1 \leq \frac{c^2 \lambda_2 s_d}{\phi_{min}^2}$ and use of the bound prescribed gives $l_d u_1 = o(1)$.

$\square$

# B   Additional Experimental Details

We show the corrected reproducibility results in Figure 6. For multiple testing correction in our experiments We use the Benjamin-Hochberg FDR procedure.

**Figure 6:** Reproducibility of results from sub-sampling using FDR of 5% Reproducibility of results from subsampling, debiased lasso does not produce any significant edge differences that correspond to a 5% error rate