[Reviews · NeurIPS 2016]

Reviewer 1

Summary

This paper proposes a new methodology to learn the difference of brain connectivity between different populations, based on Gaussian Graphical Models and fMRI data. One of the key aspects of this application is that the connectivity matrix of a brain is NOT sparse, but the deviation of connectivity between different brains IS expected to be sparse. The authors use this fact to develop a multi-task fused lasso procedure that is shown to outperform more traditional methods in detecting such differences. The framework is solid and the method is supported by thorough validation.

Qualitative Assessment

Analyzing the differences between different brains is an important application that can help neuroscientists make progress to better understand pathologies such as autism. This paper provides a customized machine learning approach for this application that significantly increases the detection of differences and the confidence in the results. As such I believe it presents a major step forward for this application that could significantly speed up knowledge discovery in the neurosciences, potentially leading to new breakthroughs in the understanding of a variety of pathologies (such as autism). The framework is solid and the method is supported by thorough validation. The paper is well written. Minor suggestions: Did you maybe forget to discuss Fig. 4 - or did I miss that? If you can find the space, it would be interesting to hear a few more details about the results shown in Fig. 5. For example, how many different brain regions did you consider (# vars)? What is thus the max. number of connections possible? How many connections did you actually detect? How sparse are the results, i.e. what is the percentage of # connections detected / # connections considered?

Confidence in this Review

2-Confident (read it all; understood it all reasonably well)


Reviewer 2

Summary

The manuscript introduces debiased multi-task fused lasso that uses sparseness constraints to improve detection thresholds for edges that are different between two graphs. The methods expands on the previous work where sparseness constrains were applied to the graph itself and now only differences are assumed to be sparse. The method is tested in simulations and applied to data comprasing resting state activity in normal and autistic patients.

Qualitative Assessment

The manuscript is clearly written, contains analytical results, numerical tests on simulated data and application to the real fmri data. The results indicate additional differences between autistic and normal patients in terms of resting state fmri signals.

Confidence in this Review

2-Confident (read it all; understood it all reasonably well)


Reviewer 3

Summary

The authors develop a method to estimate the difference in models, where models are not sparse, but the difference between models is sparse (when the models are gaussian graphical models), called "the debiased multi-task fused lasso". They discuss the application of their technique to estimating differences in functional connectivity between patient (e.g. autism) and non-patient populations. The authors use proper optimization techniques: cross-validation of hyper parameters, demonstration of the technique on synthetic data and comparison to other modern techniques.

Qualitative Assessment

It is often the case in neuroscience (and other life science disciplines) that it is much easier to collect "control" data than patient data or perturbation data (from pharmacology or optogenetics). The authors cleverly exploit an assumption in most data sets: despite having a complex model to fit controls and patients, we generally think the difference between the groups is small compared to the complexity of the model. By exploiting that feature they develop a technique to fit the difference between models rather than separately fitting both models and comparing the difference. I am a "user" rather than a developer of statistical methods, so it is hard for me to say whether this is truly novel, but I am not aware of existing literature the exploits this idea (estimating the difference), and as a neuroscientist I find the idea extremely appealing, and would like to try it in my own research.

Confidence in this Review

2-Confident (read it all; understood it all reasonably well)


Reviewer 4

Summary

After the rebuttal: I thank the authors for answering my many questions. There are important technical/experimental details left to the rebuttal that should have been in the submitted manuscript (and will hopefully be included in the final appendix). Also, I am not as optimistic as the authors about the ease of analysis if one replaces the individual sparsity assumption with another norm, but I don't think it is a major issue. I suggest that the authors modify the final text appropriately to clarify this detail. Overall, I still think this is a technically solid paper with some potential to impact the field. I have updated my scores slightly to reflect this. ===================================== The authors propose to improve a parametric (asymptotic) test for differences between precision matrices by replacing the de-biased lasso with a "multitask fused lasso". The proposed regularizer is constructed specifically for the case where the differences between the two precision matrices are sparse, in addition to separate sparsity of each precision matrices. The primary innovation is the (asymptotic) de-biasing of the estimator, resulting in a parametric (asymptotically normal) test. The technique is motivated by applications to testing differences in brain connectivity between two populations. Results on simulated data suggest that the proposed de-biasing improves statistical power to detect differences. Result on real data are focused on comparison between the proposed parametric test and the permutation based standard, suggesting similar performance without the expense of permutation.

Qualitative Assessment

The goal of improving testing for differences between graphs is clearly relevant to neuroimaging and other application domains. While the specifics are somewhat incremental, I think this is a great idea, and reasonably well executed. Major issues: * Please explain specifically which gradients are used to get from (4) to (5). This derivation seems incorrect if one takes separate derivatives with respect to \beta_1 and \beta_2. How do you end up with a sum of terms (and not two separate terms)? Or is this the sum of the two gradient terms? If accepted, I strongly suggest more a detailed derivation in the appendix. * While the authors spend significant effort motivating the need to test sparse differences between "dense" graphs (e.g. repeatedly referring to [19] to suggest that brain networks are not sparse), the resulting techniques are in-fact designed to test sparse differences between "sparse" graphs. I think this is unfortunate, because the idea of improving tests for edge differences is sufficiently interesting and potentially useful without the mismatch between motivation and algorithm. I strongly suggest that the authors modify the motivation to better reflect the proposed test. * One of the issues with the regression approach to precision matrix estimation is that one gives up finite sample guarantees of both symmetry and positive semi-definiteness. Please add some details on how you handle this e.g. do you symmetrize before testing for differences? How do you do this? Do you implement cross-validation using the log likelihood? If so, how do you implement the determinant? I think these details are important for reproducibility and practical utility. * I am not convinced of the motivation to (significantly) over-sample one of the groups in the two group test e.g. for simulations n1=800, n2=20 - 150. The authors suggest that this reflects of practice, but I am not aware of many situations where this occurs. Instead, most datasets collected for two sample testing tend to be balanced. Could the authors add more detail or a reference? * On a related note, Could the authors comment on how the results change if the two groups are balanced? Is there still a gain from using the fused Lasso? How much? * The algorithms are designed for individual sample level data, while the presented experiments are at the group level. As the fMRI data is collected for each individual, how do the authors construct samples used in the presented experiments? Are the data concatenated for all subjects in each group? Please clarify. Minor issues: * Please note the number of brain regions used for the fMRI experiments. This is an important detail that is easy to add. Suggestions: * It would be quite informative to compare to "multitask fused lasso" results with no debasing at all, to clarify what is gained.

Confidence in this Review

2-Confident (read it all; understood it all reasonably well)


Reviewer 5

Summary

This paper introduced a so-called de-biased multi-task fused lasso to characterize the differences between two sparse GGM networks. The author demonstrated the proposed method on one synthetic dataset and one real data (ABIDE Brain Imaging repo).

Qualitative Assessment

This paper extends the debiased lasso estimator to the neighborhood based structure estimation for GGM. Then the author used the debiased multi-task fused lasso to estimate the differential structure between two GGMs. Though the formulation is interesting, the paper has a few critical issues: 1. baselines In the experiments, the proposed fused lasso was only compared to two neighborhood based structure estimators: (1) debiased lasso (2) hdi. They are certainly not the state-of-art baselines. For instance, the paper should at least compare with JGL that also uses the fused lasso penalty .. (just use JGL to get the two graphs, and then substract to get the difference). It is hard to tell the benefits of the proposed versus JGL-fused without this comparison. More accurate ? More efficient ? 2. references Connected to the baseline issue, the paper missed a few important references, e.g., - Generalized Direct Change Estimation in Ising Model Structure Farideh Fazayeli University of Minnesota, Arindam Banerjee - Structure Learning of Partitioned Markov Networks Song Liu The Inst. of Stats. Math., Taiji Suzuki , Masashi Sugiyama University of Tokyo, Kenji Fukumizu The Institute of Statistical Mathematics - The Multiple Quantile Graphical Model, A Ali, JZ Kolter, RJ Tibshirani - Identifying gene regulatory network rewiring using latent differential graphical models 3. the sparse assumption The authors claimed (e.g. on line 59) that functional networks are not sparse. But in the proposed debiased multi-task fused lasso, each graph's parameter was constrained with the L1 penalty. They seem contradictory ? 4. significance calculation One of the major contribution this paper claimed was about confidence interval and p-value calculation to evaluate the uncertainty of derived GGM network differences. However, the paper includes almost no detailed formula or description about how this should be performed (only in line 220-221 ??). 5. Minor issue the paper includes quite some wording mistakes, e.g., - line 189: bounds gives - line 193: messy equation references - line 203: notes this leads ? - line 160: in addition these ...

Confidence in this Review

2-Confident (read it all; understood it all reasonably well)